# Directional Gaussian Mixture Models of the Gut Microbiome Elucidate Microbial Spatial Structure

Amey P. Pasarkar,[a] Tyler A. Joseph,[a] (ID) Itsik Pe'er[a,b,c]

[a]Department of Computer Science, Columbia University, New York, New York, USA
[b]Department of Systems Biology, Columbia University, New York, New York, USA
[c]Data Science Institute, Columbia University, New York, New York, USA

**ABSTRACT** The gut microbiome is spatially heterogeneous, with environmental niches contributing to the distribution and composition of microbial populations. A recently developed mapping technology, MaPS-seq, aims to characterize the spatial organization of the gut microbiome by providing data about local microbial populations. However, information about the global arrangement of these populations is lost by MaPS-seq. To address this, we propose a class of Gaussian mixture models (GMM) with spatial dependencies between mixture components in order to computationally recover the relative spatial arrangement of microbial communities. We demonstrate on synthetic data that our spatial models can identify global spatial dynamics, accurately cluster data, and improve parameter inference over a naive GMM. We applied our model to three MaPS-seq data sets taken from various regions of the mouse intestine. On cecal and distal colon data sets, we find our model accurately recapitulates known spatial behaviors of the gut microbiome, including compositional differences between mucus and lumen-associated populations. Our model also seems to capture the role of a pH gradient on microbial populations in the mouse ileum and proposes new behaviors as well.

**IMPORTANCE** The spatial arrangement of the microbes in the gut microbiome is a defining characteristic of its behavior. Various experimental studies have attempted to provide glimpses into the mechanisms that contribute to microbial arrangements. However, many of these descriptions are qualitative. We developed a computational method that takes microbial spatial data and learns many of the experimentally validated spatial factors. We can then use our model to propose previously unknown spatial behaviors. Our results demonstrate that the gut microbiome, while exceptionally large, has predictable spatial patterns that can be used to help us understand its role in health and disease.

**KEYWORDS** Gaussian process, MaPS-seq, computational biology, machine learning, mathematical modeling, microbiome, probabilistic models, spatial structure

A defining characteristic of the gut microbiome community is its spatial structure. Nutrients and chemical conditions differ along the gastrointestinal (GI) tract, impacting the distribution of taxa that reside there (1, 2). This spatial arrangement of microbes within the gut microbiome likely contributes to major aspects of its dynamic behavior, including community stability and host-microbe interactions (3, 4).

Recently, a novel DNA technology, metagenomic plot sampling by sequencing (MaPS-seq), was developed to offer insights into the spatial organization of the gut microbiome (5). In MaPS-seq, high-resolution segments ($\sim$20-$\mu$m squares) are extracted directly from along the gut. Segments are encapsulated in droplets with barcoded 16S rRNA amplification primers, such that sequencing reads with the same barcode originate from the same segment. Hence, MaPS-seq preserves localized information about the

Address correspondence to Itsik Pe'er, itsik@cs.columbia.edu.

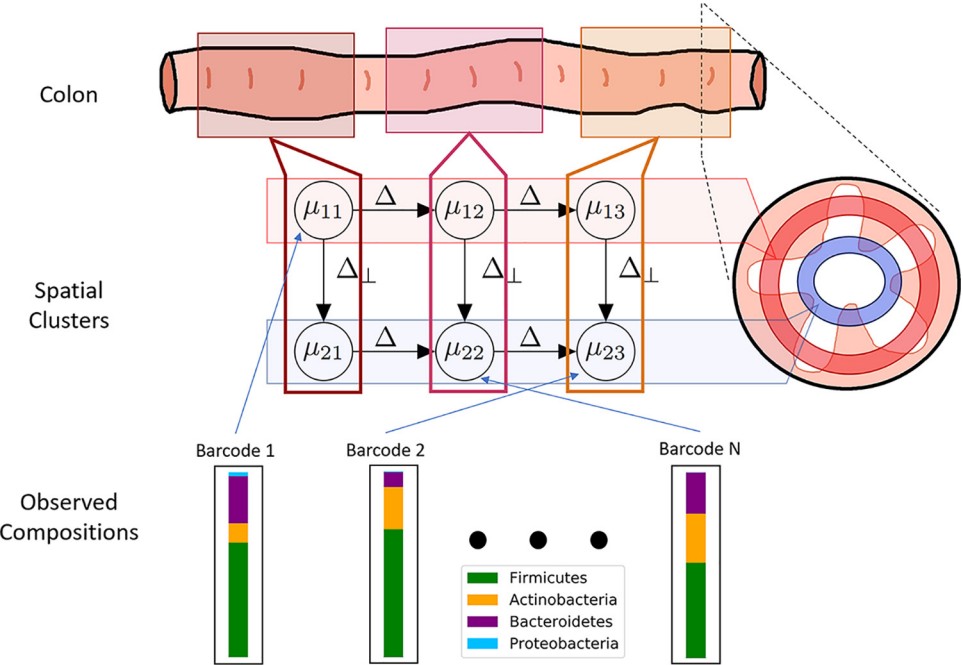

**FIG 1** Schematic overview of the directional Gaussian mixture model. Given observed compositions from each barcode, the model simultaneously learns the community composition of each latent cluster ($\mu_i$) and the assignment of each barcode to a latent cluster.

spatial structure of the microbiome and is a valuable tool for investigating the biogeography of the gut microbiome. However, the assignment of barcodes to droplets is a random process; MaPS-seq does not preserve the global arrangement of droplets along the gut.

Known characteristics of the biogeography of the gut microbiome suggest it may be possible to reconstruct the global arrangement of MaPS-seq droplets. For example, antimicrobial peptides, oxygen levels, and acidity vary along the length of the small intestine. Consequently, bacterial loads increase along the longitudinal axis of the small intestine and lead to a more microbe-rich ileum (2). In the colon, the density of the mucus layer increases along its longitudinal and cross-sectional axes—creating environmental niches favored by different species (1). In principle, it should be possible to reconstruct some of these global patterns from the high-resolution sampling of MaPS-seq.

**Our contribution.** We developed a class of computational models to recover known characteristics of the biogeography of the gut microbiome from MaPS-seq data. Our models build upon the classical Gaussian mixture model (GMM; Fig. 1). In a GMM, observations are mixtures of latent clusters, each of which is modeled as a multidimensional Gaussian random variable, independent of the others and with its own mean. We expand this framework by introducing spatial dependence between latent clusters. Specifically, clusters are arranged as a line (one-dimensional model) or grid (two-dimensional model) to investigate directional changes along the longitudinal axis only or, respectively, both the longitudinal and radial axes of the gut.

A key question is whether our model can differentiate longitudinal from radial changes in the gut. We demonstrate on synthetic data that our model is capable of discriminating between one-dimensional and two-dimensional models. We apply our model to MaPS-seq mouse ileum, cecal, and distal colon data sets. We provide strong evidence for the presence of spatial structures across all data sets, with distinct regional characteristics. We show that our proposed model recovers known biological behaviors of microbes within the GI tract while also providing new insights into the spatial structure of the gut microbiome.

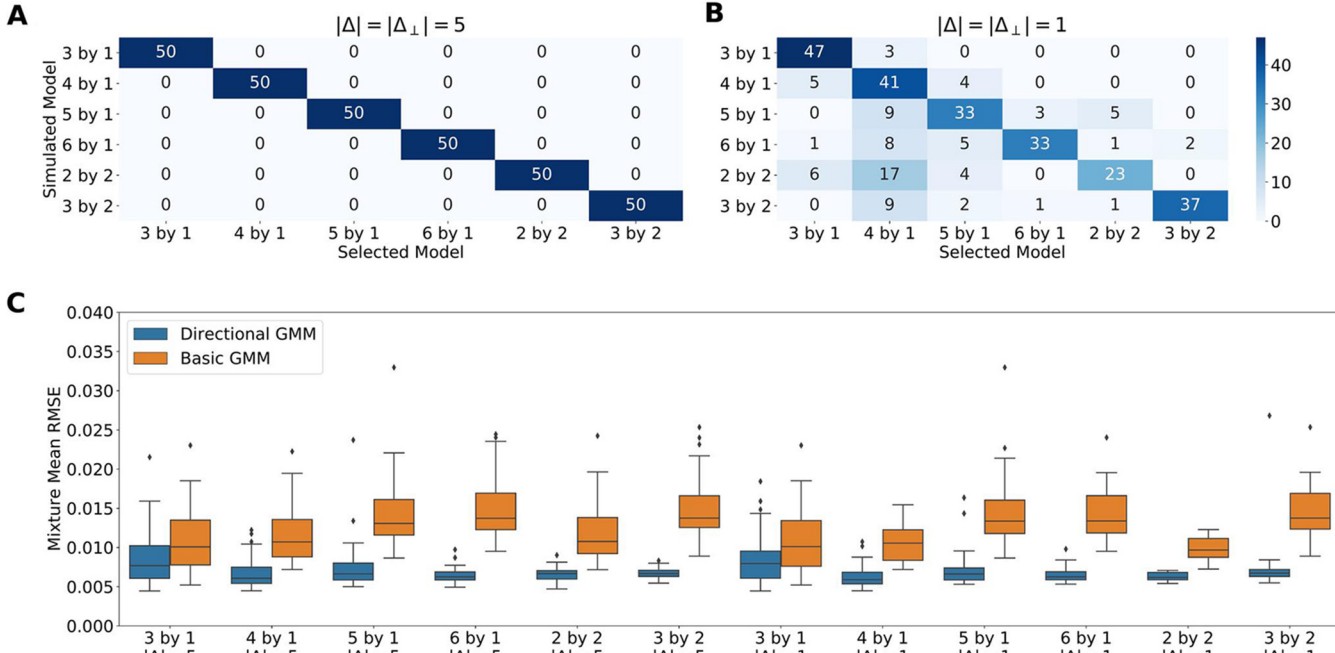

**FIG 2** Directional GMMs accurately select the number of latent clusters and infer model parameters. (A) Heatmap showing the accuracy of the selected model on various simulated data sets with $|\Delta| = |\Delta_\perp| = 5$ and a within-cluster standard deviation of 1. (B) $|\Delta| = |\Delta_\perp| = 1$, keeping the same standard deviation. (C) RMSE of learned cluster means on data sets with correct model selection. On all data sets, the directional GMMs are significantly improving parameter inference.

## RESULTS

**Simulation results.** We first evaluated whether our model can differentiate between one- and two-dimensional dynamics using simulated data. We simulated data under the one- and two-dimensional models ("Simulation Analysis," below) and asked if we could infer the number of latent clusters and their spatial arrangement. One-dimensional models use a $\Delta$ vector to describe changes between latent clusters arranged in a line. Two-dimensional models have latent clusters arranged in a grid and use an additional $\Delta_\perp$ vector to describe orthogonal changes ($\Delta \cdot \Delta_\perp = 0$). Using the Akaike information criterion (AIC), we found that our directional GMM is able to correctly determine the correct number and arrangement of clusters (Fig. 2A and B). Furthermore, the introduction of a dependence between latent clusters in the model also improved parameter inference compared to a naive GMM with no spatial structure (Fig. 2C). For all data set forms, the Wilcoxon signed-rank test $P$ value was less than 0.001.

**Spatial structure of MaPS-seq data.** We applied our directional GMM to three real MaPS-seq data sets from Sheth et al. (5; "MaPS-seq data analysis," below). The provided MaPS-seq data contain samples from 3 regions of a single mouse's GI tract, the cecum ($n = 4.5$ barcodes), the ileum ($n = 386$ barcodes), and the distal colon ($n = 259$ barcodes). On the cecum and distal colon data sets, the best-supported models were two-dimensional ($4 \times 2$ and $3 \times 2$, respectively). On the ileum data set, the best-supported model was a one-dimensional model with 5 clusters ($5 \times 1$). Using the model parameters from the best-supported model on each data set, we created one- and two-dimensional visualizations depicting the directions learned by our model (Fig. 3). Qualitatively, our model appeared to segregate barcodes into distinct clusters along the gut.

We also compared the support of the selected directional model GMM to a naive GMM with no spatial structure. To compare models, we computed the AIC scores of our directional models to a naive GMM with the same number of latent clusters (Table 1). The naive GMMs have much larger AIC scores than the directional GMMs. Conventionally, models with scores that are larger by 10 or more are considered to have little support (6).

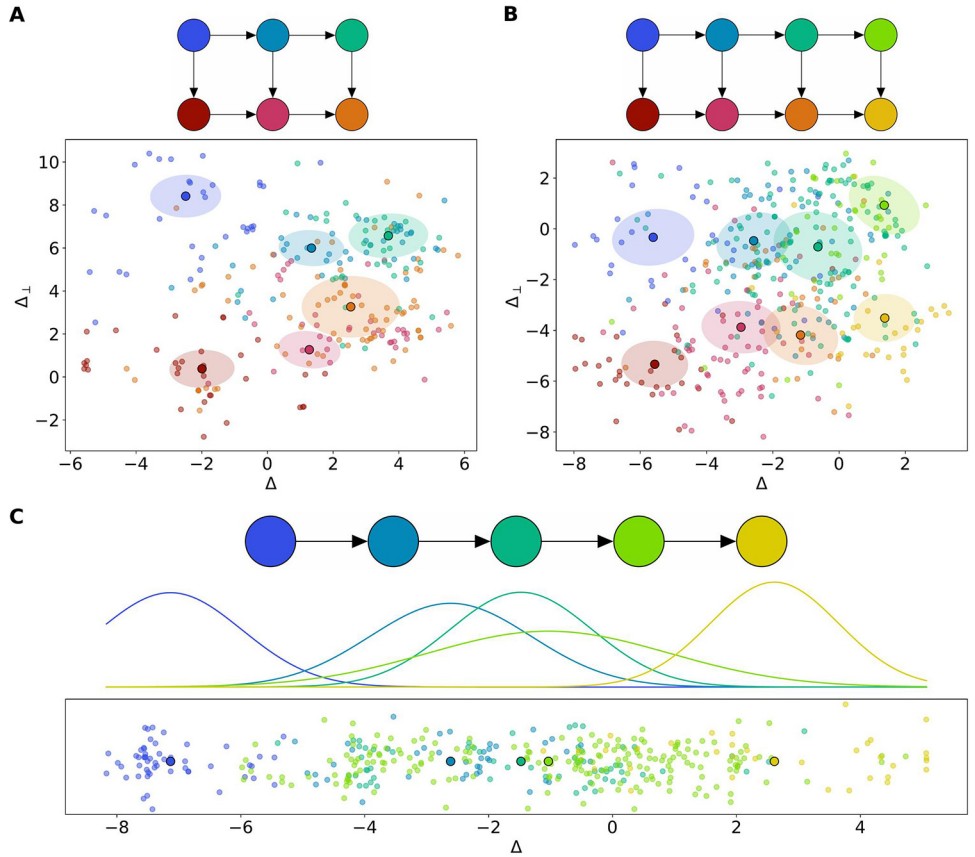

**FIG 3** Projections of MaPS-seq data. (A) Distal colon data set. The selected model and MaPS-seq data projected along the unit $\Delta$ and $\Delta_\perp$ axes. Colors correspond to samples belonging to a latent cluster. Ellipse radii represent the eigenvectors of the covariance matrix. (B) Cecum data set. (C) Ileum data set. Selected model and MaPS-seq data are projected along the unit $\Delta$ axis. Normal distribution represents the density of covariance matrices around each cluster mean.

**Recovery of GI tract biogeography.** We also investigated learned model parameters for correspondence to some of the known spatial dynamics of the gut microbiome. Our mixture models identify latent clusters, each of which describes a community state. We can analyze the average composition of these latent clusters to identify variations in microbial compositions between locations of the GI tract.

Figure 4 illustrates the recovered dynamics on the distal colon data set. Under the partition presented in Fig. 4, we observe large differences in the average compositions of *Firmicutes* and *Bacteroidetes* between lumen- and mucus-associated clusters.

On the cecum data set, we observed compositional differences along both axes. Figure 5A to C shows a cecal tip and base partition that has a noticeable compositional difference in the abundances of *Actinobacteria* and *Bacteroidetes*. The clusters on the

**TABLE 1** AIC shows strong evidence for spatial structure across the GI tract[a]

| AIC selections | | | | | |
| --- | --- | --- | --- | --- | --- |
| | **One-dimensional** | | **Two-dimensional** | | **AIC$_{naive}$-AIC$_{structure}$** |
| **Dataset** | **Score** | **Mixture model** | **Score** | **Mixture model** | **(score difference)** |
| Ileum | **−37,493** | **5 × 1** | −37,340 | 3 × 2 | 954 |
| Cecum | 32,714 | 8 × 1 | **31,930** | **4 × 2** | 3,832 |
| Distal colon | 6,636 | **6 × 1** | **5,588** | **3 × 2** | 1,168 |

[a]Best directional mixture model and its corresponding AIC score. Scores in bold indicate the selected model. The comparison of directional GMMs to naive GMM by AIC metrics shows that the introduction of dependence between latent clusters significantly improves the model fit.

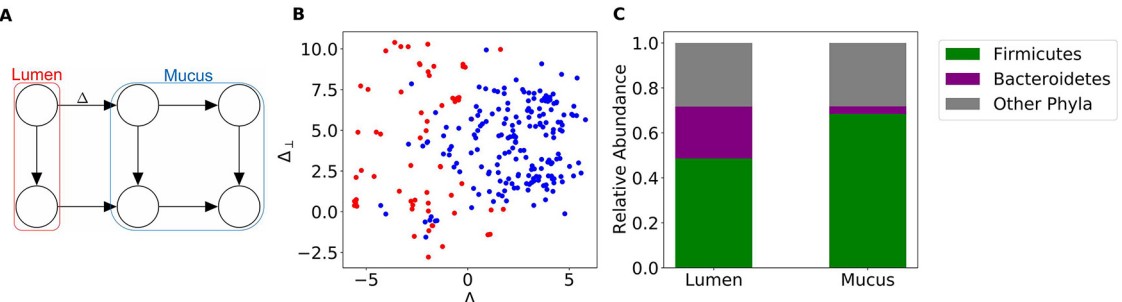

**FIG 4** Directional GMM recovers spatial dynamics in the distal colon. (A) Selected model and corresponding locations of clusters in the distal colon. (B) Scatterplot of projected MaPS-seq samples assigned to lumen-associated clusters (red) and mucus-associated clusters (blue). (C) Clusters associated with the mucus are enriched in *Firmicutes*, and those associated with the lumen display larger levels of *Bacteroidetes*.

two ends of the model have differences in the abundances of *Firmicutes* that correspond to the cecal crypt and lumen (Fig. 5D to F).

On the ileum data set, we compared microbial population relative abundances across each latent cluster. Along the length of the ileum, we observed a general decreasing trend in *Lactobacillaceae* and increases in both *Ruminococcaceae* and *Lachnospiraceae* (Fig. 6B). Our model's choice of Δ seems to capture some of these dynamics. Moving along the Δ axis shows decreases in *Lactobacillaceae* and increases in *Lachnospiraceae* (Fig. 6A). Some discrepancy is observed, most noticeably with the behavior of *Actinobacteria*.

We evaluate if the proposed latent cluster compositions still accurately describe the data. Specifically, we compare the log-likelihood of the MaPS-seq droplets in their assigned clusters against the best log-likelihood from the other clusters. These likelihoods are determined using the proposed compositions. Using a paired one-sided *t* test to evaluate the difference, we obtain $P = 0.053$.

We also calculate the Kullback-Leibler (KL) divergence between projected and observed operational taxonomic unit (OTU) abundances on each ileal segment. The divergences are

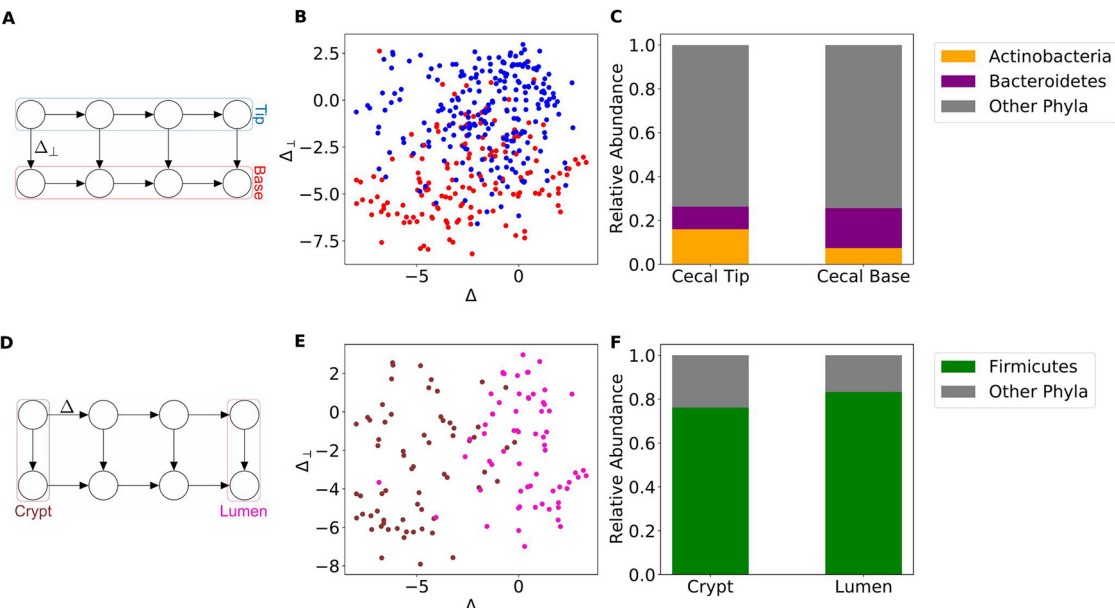

**FIG 5** Directional GMM recovers spatial dynamics in the cecum. (A) Selected model and corresponding locations of latent clusters in the cecum. (B) Scatterplot of projected MaPS-seq samples assigned to cecal tip-associated clusters (blue) and cecal base-associated clusters (red). (C) Clusters associated with the cecal tip have lower relative abundances of *Bacteroidetes* and higher relative abundances of *Actinobacteria* than the cecal base. (D) Selected model and corresponding locations of mixtures in the cecum. (E) Scatterplot of projected MaPS-seq samples assigned to cecal crypt-associated clusters (brown) and cecal lumen-associated clusters (pink). (F) *Firmicutes* are enriched in the lumen clusters compared to the crypt clusters.

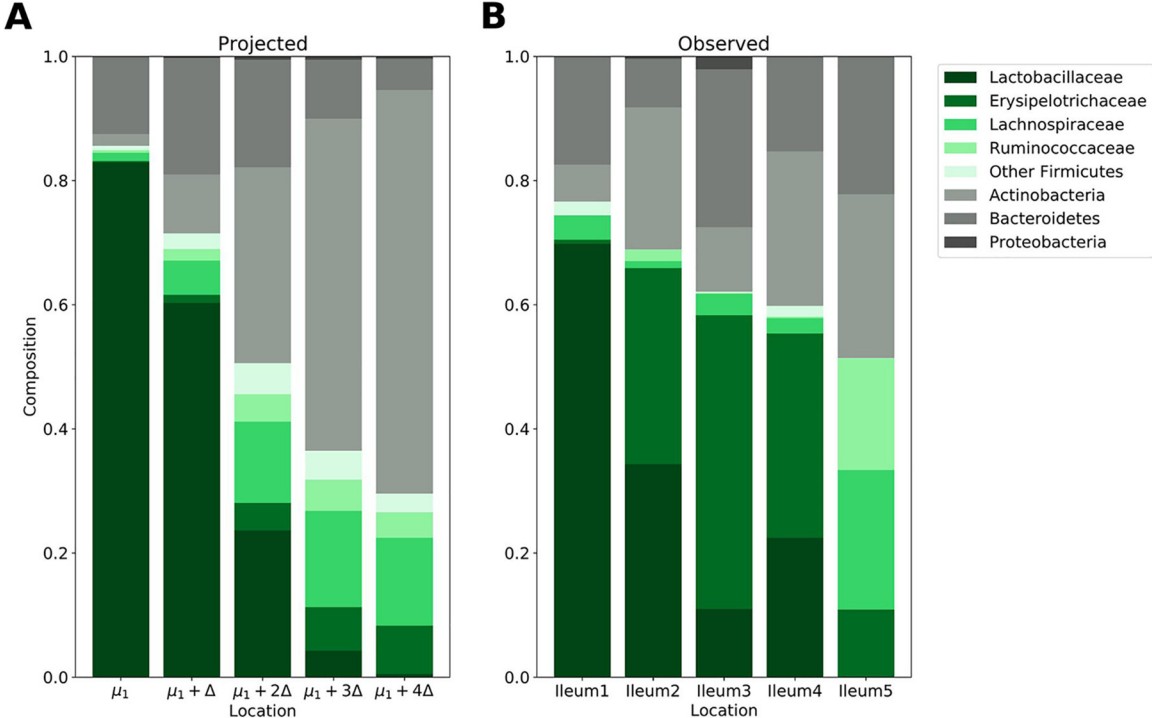

**FIG 6** Projected and observed ileum dynamics. (A) Projected compositions moving along the Δ axis. (B) Observed compositions in learned model clusters. Green bars correspond to observed families in the *Firmicutes* phylum, and gray bars correspond to other observed phyla.

0.11, 0.74, 0.96, 1.08, and 0.65 for segments 1 to 5, respectively. On all segments, *P* values in permutation tests were <0.001.

## DISCUSSION

Novel experimental methods focused on the gut microbiome's spatial organization have provided new data sets for computational analysis. Here, we developed directional GMMs with dependent mixtures to infer spatial behaviors of phyla within the gut microbiome. We demonstrated the accuracy of the proposed directional GMMs on simulated data in terms of ability to infer model parameters and to differentiate one-dimensional from two-dimensional spatial structures. On MaPS-seq data, we demonstrated the presence of spatial structure in distinct regions of the mouse GI tract. Encouragingly, our model recapitulated well-known spatial phenomena on the distal colon and cecum data sets.

In the distal colon, it has been shown that *Bacteroidetes* are enriched in the lumen, while *Firmicutes* are enriched in the mucus layer and crypts (1, 2). We observed these compositional differences suggesting that our model is recovering the radial dynamics of the distal colon. The presence of four distinct clusters representing the mucus layer is not surprising, because mucosal communities vary significantly over lengths as small as 1 cm (7).

In the cecum data set, correspondence with other *in vivo* experiments suggests that we recover dynamics in both the radial and longitudinal directions (Fig. 5A). Zaborin et al. (8) suggested that in the mouse cecum, *Bacteroidetes* increase in relative abundance from the cecal tip to the base. It should be noted that in their experiment, this trend did not reach statistical significance. Our model seems to identify this compositional difference, in addition to a distinction in the relative abundances of *Actinobacteria* (Fig. 5A). Zaborin et al. (8) did find a statistically significant difference between the levels of *Firmicutes* in the lumen compared to cecal crypts. We observe a similar difference at the two ends of our model (Fig. 5D).

Within the ileum, we select a model with only a single direction of change. There is evidence that our choice of model is biologically accurate: unlike in the cecum and distal colon, the small intestine mucus layer is largely uninhabited due to the presence of antimicrobial peptides (9). However, along the length of the small intestine, oxygen concentrations and pH gradients vary (2). Among the learned clusters, we observe a stark decrease in the relative abundance of *Lactobacillaceae* (Fig. 6). Along the flow of the digesta, the ileum becomes more alkaline. Because *Lactobacillaceae* are known to contribute to highly acidic environments, it is unsurprising that we observe these compositional differences along the length of the ileum. The pH gradient seems to be embedded in the $\Delta$ our model learns; cluster means along the $\Delta$ axis show a decrease in *Lactobacillaceae* similar to the observed compositions. The presence of discrepancies on phyla such as *Actinobacteria* suggest that there potentially exist other sources of microbial dynamics in the ileum as well. We test the scale of these divergences using two methods. First, we compare the likelihood of data based on their assigned clusters and the proposed ileum compositions against their likelihood on other clusters. A *t* test obtains a *P* value close to the threshold of statistical significance, even though this is a conservative test; we are comparing against an optimal other assignment as opposed to a random one. Second, we calculate the KL divergence between observed and proposed cluster compositions. The divergences are small, and a permutation test yields statistical significance of the divergences ($P < 0.001$). To our knowledge, there are not any experimental studies that describe the microbiome's spatial dynamics within the ileum. This demonstrates the utility of our model; not only can we computationally confirm known aspects of the gut biogeography, but we can also propose new microbial spatial behaviors.

A limitation of the present approach is the resolution of the resulting clusters. Our directional GMM was able to capture global spatial patterns in the gut microbiome.

Specifically, given that MaPS-seq samples are approximately 20 $\mu$m apart, the clusters from the best-supported models on the ileum, distal colon, and cecum data sets correspond to approximately 1-cm regions. It would be interesting to investigate if a finer resolution change can be achieved. Future work should focus on investigating this possibility of high-resolution mapping of MaPS-seq samples.

A trend in understanding microbial dynamics is uncovering microbe-microbe interactions from coabundance patterns (10). We believe that we can create more accurate interaction networks using our proposed labels of spatial locations. As certain microbe-microbe interactions may be more pronounced within certain regions of the GI tract (5), localized interaction networks can potentially uncover new microbial behaviors.

A valuable next step would be designing MaPS-seq experiments with ground truth labels denoting spatial locations. With coarse-grained labels from various adjacent segments of the GI tract, we could better confirm our model's ability to identify the microbiome's spatial structure, and also the spatial scale recovered by the model. Nonetheless, the present work provides strong evidence that global spatial patterns can be reconstructed from MaPS-seq data that will only be improved with more detailed collection.

## MATERIALS AND METHODS

**Directional Gaussian mixture models.** Our approach uses a Bayesian network to describe the relationship of spatially arranged clusters in the gut (Fig. 7A). In detail, given a spatial configuration, the goal is to simultaneously learn community states for each latent cluster and assign barcoded MaPS-seq droplets to a cluster (Fig. 1). The nodes of the Bayesian network, $\{\mu_s | s \in S\}$, represent composition vectors of archetypal communities in respective clusters.

In the present work, we are interested in changes along one or two dimensions. Studies suggest the presence of two natural directions in the gut microbiome (1). One dimension moves along the flow of the digesta, while the other moves orthogonally along the radial axis (inward out).

This motivates the following definition for our model.

We define a one-dimensional model where $S = \{i | 1 \leq i \leq K\}$ for $K$ latent clusters (Fig. 7A). Let $\Delta$ represent directional changes between adjacent community compositions and let $Q$ represent a spatial covariance matrix for deviations from the directional changes. $\overline{\mu}_0$ and $Q_0$ serve as a starting multivariate cluster that we draw the first latent cluster from.

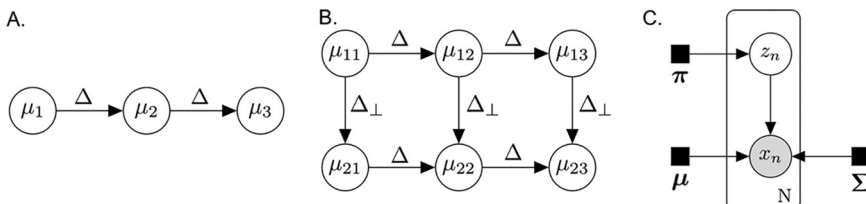

**FIG 7** A directional Gaussian mixture model. (A) Graphical depiction of relationships between latent clusters in a one-dimensional model. (B) Relationships in a two-dimensional model, where changes from left-to-right are described by $\Delta$, and perpendicular changes are described by $\Delta_\perp$. (C) GMM used to model sampling noise of observed samples $x_n$. $\mu_i$ represents latent clusters with relationships given in panels A and B.

We can define

$$f(\mu_1) = \mathcal{N}\left(\mu_1|\overline{\mu}_0, Q_0\right)$$

$$f(\mu_i|\mu_{i-1}) = \mathcal{N}\left(\mu_i|\mu_{i-1} + \Delta, Q\right) \quad \text{for } i = 2..K$$

We also define a two-dimensional model where $S = \{(i, j)|1 \leq i \leq K, 1 \leq j \leq 2$ for $2K$ latent clusters (Fig. 7B). Let $\Delta_\perp$ represent the direction along the second dimension, such that $\Delta \cdot \Delta_\perp = 0$. We define

$$f(\mu_{11}) = \mathcal{N}\left(\mu_{11}|\overline{\mu}_0, Q_0\right)$$

$$f(\mu_{1i}|\mu_{1(i-1)}) = \mathcal{N}\left(\mu_{1i}|\mu_{1(i-1)} + \Delta, Q\right) \quad \text{for } i = 2..K$$

$$f(\mu_{21}) = \mathcal{N}(\mu_{21}|\mu_{11} + \Delta_\perp, Q)$$

$$f(\mu_{2i}|\mu_{1i}, \mu_{2(i-1)}) = \mathcal{N}\left(\mu_{2i}|\frac{1}{2}(\mu_{1i} + \mu_{2(i-1)} + \Delta + \Delta_\perp), Q\right) \quad \text{for } i = 2..K$$

MaPS-seq outputs read counts for each of the operational taxonomic units (OTUs) in each barcoded droplet. However, the total number of reads is independent of overall community size. Therefore, the sequencing counts only provide information about the proportions of each OTU in the community. Recent work has advocated using such compositional data transformations to model microbiome data (11). We transformed read counts to relative abundances and then applied the PhILR transformation, an isometric log-transform (ILR) with a phylogenetically derived basis (12). Each coordinate for the PhILR-transformed data measures the relative proportions of two clades in a phylogeny. Phylogenetic trees were generated using QIIME (13) and provided as input to the PhILR R package. Given $D$ taxa, the latent community states are $D-1$ dimensional vectors, $\mu_s \in \mathbb{R}^{D-1}$. Zeros are handled using multiplicative replacement with $\delta = 1/D^2$ for $D$-taxa (14).

The reads in a particular barcoded droplet provide noisy observations from a latent cluster (Fig. 7C). Thus, we can think of the data generation process as first selecting a latent community state per barcode and then generating a noisy observation from that community state. Let $\mathcal{B}$ index the set of barcodes, $x_b \in \mathbb{R}^{D-1}$ for $b \in \mathcal{B}$ be PhILR computed from the observed sequencing reads for that barcode, and $\pi_S = (\pi_s)_{s \in S}$ be the probability that a barcode originated from each cluster $s \in S$. Let $\rho_s$ be the set of direct ancestors of $\mu_s$. Defining $z_b$ as the cluster from which barcode $b$ originated, we have

$$p(z_b) = \text{Categorical}\,(z_b|\pi_S)$$

$$p(x_b|z_b, \mu_{z_b}) = \mathcal{N}(x_b|\mu_{z_b}, \Sigma_b) = \prod_{s \in S}\left[\mathcal{N}(x_b|\mu_s, \Sigma_s)\right]^{1(z_b=s)}$$

Altogether, the complete likelihood of the model can be written

$$p(\mu_S, z_\mathcal{B}, x_\mathcal{B}) = \prod_{s \in S} f(\mu_s|\mu_{\rho_s}) \prod_{b \in \mathcal{B}} p(x_b|z_b, \mu_{z_b}) p(z_b)$$

$$= \prod_{s \in S} f(\mu_s|\mu_{\rho_s}) \prod_{b \in \mathcal{R}} \prod_{s \in S} \left[p(x_b|z_b = s, \mu_s) p(z_b = s)\right]^{1(z_b=s)}$$

**Parameter inference.** In both models, we seek to optimize $p(\mu_S, z_\mathcal{B}, x_\mathcal{B}|\theta)$, where $\theta = (\pi_S, \sum_S, \Delta_\cdot, Q, Q_0, \mu_0)$. This optimization is performed through an expectation-maximization (EM) algorithm. Under this algorithm, parameters are inferred by alternating between two steps:

- **E step:** Given the current estimates of community states $\mu_S^t$, model parameters, $\theta^t$, compute the posterior expectation of each cluster assignment, $[1(z_b = s)|\mu_S^t, \theta^t]$.
- **M step:** Maximize the expected complete log-likelihood $\log p(\mu_S, z_{\mathscr{B}}, x_{\mathscr{B}})$:

$$\left(\mu_S^{t+1}, \theta^{t+1}\right) = \operatorname{argmax}_{(\mu_S, \theta)} \sum_{s \in \mathcal{S}} \log f\left(\mu_s | \mu_{\rho_s}\right)$$

$$+ \sum_{b \in \mathcal{B}} \sum_{s \in \mathcal{S}} \left[1(z_b = s)|\mu_S^t, \theta^t\right] \left[\log p(x_b|z_b = s, \mu_s) + \log p(z_b = s)\right]$$

Thus, we take maximum *a posteriori* estimates of $\mu_s$ and maximum likelihood estimates of the remaining parameters. Model parameters are initialized using a basic GMM with independent clusters trained on the same data. On both simulated and real data, 200 initializations are used. Inference terminates following 5 consecutive steps, where the expected complete log-likelihood increases relatively by $<10^{-4}$ of the previous step. An initialization is restarted if it does not converge within 500 iterations.

**Model selection.** The Akaike information criterion (AIC) was used to evaluate models.

$$\text{AIC}(k) = -2\ln(\hat{L}) + 2p_k$$

where $\hat{L}$ denotes the likelihood of the data under the fitted model and $p_k$ is the number of parameters for the model $k$. We used the complete log likelihood as a surrogate for the log likelihood of the data since it is a lower bound. When comparing models with different numbers of latent clusters, we choose the model with the minimum AIC score (6).

In the case of models with the same number of latent clusters (i.e., 4 clusters arranged in a line versus clusters arranged in a 2 by 2 grid), we can directly compare the complete likelihoods $p(\mu_S, z_{\mathscr{B}}, x_{\mathscr{B}})$ of either model.

On both simulated and real data, we test up to 8 clusters for the one- and two-dimensional models, or until the average community state size is 50 samples, whichever comes first. For the one-dimensional model, clusters were arranged in a line from a $2 \times 1$ model up to an $8 \times 1$ model. For the two-dimensional model, clusters were arranged in a grid from a $2 \times 2$ model up to a $4 \times 2$ model.

**Simulation analysis.** This is an unsupervised learning problem, so we first evaluated our model on simulated data. To this end, we create simulated data sets under the two proposed models. First, we sample two clusters' means from a Pareto distribution with $\alpha = 1$, normalize to the relative abundance space with $D = 47$ taxa, sort taxa in decreasing order, and then transform to the ILR space. The difference in the ILR space between these two means is defined to be $\Delta$. The $\Delta$ parameter is then scaled to our desired magnitude. Our method for sampling $\Delta$ allows for larger dynamics to be observed on more abundant taxa. For two-dimensional models, we sample $\Delta_\perp$ from a standard multivariate normal distribution and then orthogonalize relative to $\Delta$. The remaining cluster means are arranged around one of the two original cluster means as per the two models (arranged in the ILR space in a line or in a grid).

Then, we randomly sample cluster covariance matrices $\sum$ from an inverse-Wishart distribution with $\nu = D + 1$ and $\Psi = \frac{I_{D-1}}{D-1}$ (15). Finally, a total of 360 artificial MaPS-seq samples are drawn evenly and independently from each cluster.

We analyzed two aspects of model performance on simulated data: (i) selection of the correct number of latent clusters and (ii) parameter estimation accuracy. In order to evaluate our model selection framework, we train both the one- and two-directional models with various amounts of latent clusters on simulated data. We used the aforementioned model selection criteria to determine the optimal model.

Next, the accuracy of our parameter inference is determined by calculating the average root of the mean square error (RMSE) of the learned cluster means. Although our proposed model assigns labels to clusters to reflect their spatial arrangements, other unsupervised clustering algorithms assign arbitrary labels. Therefore, to compare RMSE of model parameters, we look at our proposed model's RMSE and the best RMSE of all label permutations of a naive GMM.

**MaPS-seq data analysis.** We used the publicly available data from Sheth et al. (5). The cecum, ileum, and distal colon data sets were each extracted from 3-cm segments of their respective regions. MaPS-seq clusters are the same size in all data sets (20 $\mu$m). Each sample is a vector of the relative abundances of all OTUs. We focused on the most abundant taxa that constitute 95% of all relative abundance across the three data sets. This corresponded to 47 taxa. Relative abundances were then renormalized. Using the provided fasta files, we generated phylogenetic trees in QIIME (13). Data were then transformed using the PhILR R package.

To determine how well proposed regions in the ileum describe the data, we compared the log-likelihood of PhILR-transformed data in their assigned cluster against the highest log-likelihood of other proposed region compositions. We used the same $\sum$ covariance matrices. A paired one-sided $t$ test was used to calculate $P$ value.

To quantify divergence between proposed and observed regions in the ileum, we used KL divergence. For each region of the ileum, we calculated the relative entropy from the observed and proposed compositions. In order to determine the significance of the calculated KL divergences, we used a permutation test with 10,000 permutations of each observed composition vector.

**Data availability.** MaPS-seq data are publicly available at https://github.com/ravisheth/mapsseq. EM algorithms and analysis of MaPS-seq data are available at https://github.com/amepas/Spatial_Mbiome.

## ACKNOWLEDGMENTS

We thank Harris Wang and Miles Richardson for sharing the data and useful conversations. Research was supported by a seed grant from the Data Science Institute at Columbia and a Driving Biological Project award from the CaST Institute (U54CA209997).

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
