## [Reviewer comments · mSystems]

Directional Gaussian Mixture Models of the gut microbiome elucidate microbial spatial structure

Amey Pasarkar, Tyler Joseph, and Itsik Pe'er

Corresponding Author(s): Itsik Pe'er, Columbia University

Review Timeline:

Submission Date:	July 7, 2021
Editorial Decision:	September 4, 2021
Revision Received:	September 28, 2021
Accepted:	October 6, 2021

Editor: Vanni Bucci

Reviewer(s): The reviewers have opted to remain anonymous.

Transaction Report:

DOI: <https://doi.org/10.1128/mSystems.00817-21>

September 4, 2021

Dr. Itsik Pe'er
500 w 120th st
500 w 120th st
mailcode 0401
mailcode 0401
New York, NY 10027

Re: mSystems00817-21 (Directional Gaussian Mixture Models of the gut microbiome elucidate microbial spatial structure)

Dear Dr. Itsik Pe'er:

Thank you for submitting your manuscript to mSystems. We have completed our review and I am pleased to inform you that, in principle, we expect to accept it for publication in mSystems. However, acceptance will not be final until you have adequately addressed the reviewer comments.

Preparing Revision Guidelines

Sincerely,

Vanni Bucci

Editor, mSystems

Journals Department
Reviewer comments:

Reviewer #1 (Comments for the Author):

In their paper Pasarkar et al., 2021 design a Bayesian network-derived Gaussian mixed model, in order to establish generative classifiers and model for the topographical biogeography of gut microbiome using MaPS-seq data.

The paper is well written and establishes a theoretical modelling structure that shows some relative similarity to microbiome biogeography. I have some minor comments:

1. Please make sure to release the github repo prior to sharing submitting your article to any journal. The current link does not re-direct to a public repo, as most probably the author (github user: amepas) has kept the repo as private. If not at least provide coding structure in the form of a txt file as supplemental information. In the revision, please ensure the link is functional.
2. Perhaps I have not understood or missed this, but why did you use only 20 initializations on simulated vs. 200 on real data? Did you reach the threshold of the log-likelihood being $<10^{-4}$? Should you not use similar initializations, does the difference reflect a lack of complexity in the simulated data and therefore preclude its need? Please explain.
3. Although you have run a comparison of the model yet somehow the projected data seems to show strong divergence from observed data. What is missing for me is formal hypothesis testing using a statistical test to evaluate Kullback-Leibler divergence of projected and observed spatial compositional data. Although you have used the AIC method to evaluate your model, a formal test is missing to compare projected and observed outcomes in real-world model deployment. There are major discrepancies in the data presented not 'some'. Not only Actinobacteria, but also Erysipelotrichaceae, Lachnospiraceae, Ruminococcaceae and in fact also Lactobacillaceae show deviation from observed data. Please explain? Please provide metrics showing how much the projected and observed data diverge, perhaps by using Pearson's residuals or another metric of multivariate comparison.

Reviewer #2 (Comments for the Author):

Pasarkar and colleagues present a method to retrieve the spatial structure of the gut microbiome based on Metagenomic Plot Sampling by sequencing (MaPS-seq) data. This technique allows to infer the microbial composition using barcoded 16S rRNA amplification of specific locations in the gastrointestinal (GI) tract, however, the specific location of each sample is lost. The authors propose a directional Gaussian Mixture Model (GMM) that has a spatial structure that mimics the locations from which samples were taken. The method is illustrated on simulated and real-world data, from which the latter stems from the original publication in which MaPS-seq was proposed (Sheth et al., 2019).

The proposed method is relevant, understandable and could be an interesting tool for future gut microbiome research.

I have some questions, mostly minor and concerning the methodology, and a few remarks:

1. How do you see your method being incorporated in future gut microbiome research? Can you maybe give one or two concrete applications for which the method might be beneficial?
2. As the fitting of the GMM is a stochastic process, how stable is the choice of one- and two-dimensional models based on the AIC? Also, the number of evaluated topologies seems rather small, especially for the two-dimensional models (from 2x2 to 2x4). Why not go higher?
3. How do you choose the covariance matrix of each mixture? Are these fully independent from each other, or are some of the coefficients shared?
4. Why do you focus on 95% of most abundant taxa over all three datasets instead per dataset?
5. How were the reads preprocessed? At which taxonomic level did you analyze them? If I am not mistaken this is the OTU level, but results for Fig. 6 are displayed on the phylum level. Did you find similar patterns at the OTU level?
6. Are the reads available on a public repository?
7. l241: What is the Inverse-Wishart distribution?
8. Did you use other software packages to implement the method? If so, please reference them.
9. Some figures have small captions, which are difficult to read in its current form. See for example the labels in Figures 4 and 5.

10. A suggestion (especially as you have quite some space left): as the Materials & Methods section comes in the end, the paper might be more accessible if you repeat some of the essential elements from it in the Results section. Some things (e.g. the simulation study) only became understandable after reading the paper as a whole. Another thing I would add is that each mixture represents a composition, from which you use the average to inspect changes in composition between different (estimated) locations in the GI.

11. The link to the repository with code (and data?) does not seem to work.

There is a typographical error and a few inconsistencies:

- l22: seem → seems
- Names of datasets are sometimes written with and sometimes without capital letter. Please be consistent.
- Please introduce the meaning of all symbols, such as Q , Q_0 , μ_0 , and z_b and upon first mentioning, such as Δ and $\Delta \perp$.

Reviewer #1 (Comments for the Author):

1. Please make sure to release the github repo prior to sharing submitting your article to any journal. The current link does not re-direct to a public repo, as most probably the author (github user: amepas) has kept the repo as private. If not at least provide coding structure in the form of a txt file as supplemental information. In the revision, please ensure the link is functional.

The github repository has been made publicly available.

2. Perhaps I have not understood or missed this, but why did you use only 20 initializations on simulated vs. 200 on real data? Did you reach the threshold of the log-likelihood being $<10^{-4}$? Should you not use similar initializations, does the difference reflect a lack of complexity in the simulated data and therefore preclude its need? Please explain.

We have modified the methodology to use 200 initializations on both the simulated and real data (lines 290-291). The corresponding results have been updated.

We have further clarified the terminating threshold used (line 294): an initialization will terminate once it reaches the convergence threshold of a relative increase in the log-likelihood being less than 10^{-4} . If this does not occur within 500 iterations, the initialization is restarted.

3. Although you have run a comparison of the model yet somehow the projected data seems to show strong divergence from observed data. What is missing for me is formal hypothesis testing using a statistical test to evaluate Kullback-Leibler divergence of projected and observed spatial compositional data. Although you have used the AIC method to evaluate your model, a formal test is missing to compare projected and observed outcomes in real-world model deployment. There are major discrepancies in the data presented not 'some'. Not only Actinobacteria, but also Erysipelotrichaceae, Lachnospiraceae, Ruminococcaceae and in fact also Lactobacillaceae show deviation from observed data. Please explain? Please provide metrics showing how much the projected and observed data diverge, perhaps by using Pearson's residuals or another metric of multivariate comparison.

The noticeable discrepancies are not surprising due to the overall complexity of the system. However, there are some general trends that are shared between projected and observed data, specifically on *Lactobacillaceae*.

To your point, we have performed two hypothesis tests of the divergence (lines 152-161). First, we determined if the proposed latent clusters still accurately describe the data. We compared the likelihood of MaPS-seq droplets in their assigned latent cluster against the highest likelihood of all other clusters (line 356). Note, that this is a conservative test, as it compares against an

optimal other assignment, rather than a random one. The likelihood was calculated using the proposed compositions and the same covariance matrices learned by the model. If the proposed compositions are extremely far from the observed compositions, then we would expect that many MaPS-seq droplets would likely have a higher likelihood for other clusters. We observe a p-value that is only barely higher than the traditional threshold for statistical significance.

We also test using the KL divergence metric. On each region of the ileum, we have calculated the KL divergence between projected and observed compositions, and an associated p-value from a permutation test (line 159). The permutations are on the ordering of OTUs in the observed data. This allows us to test if the OTUs are contributing similarly to the observed and projected compositions.

In short, we find that the projected compositions still maintain cluster assignments reasonably well and that OTU compositions are similar between projected and observed data as per the KL divergence.

Reviewer #2 (Comments for the Author):

I have some questions, mostly minor and concerning the methodology, and a few remarks:

1. How do you see your method being incorporated in future gut microbiome research? Can you maybe give one or two concrete applications for which the method might be beneficial?

We have added one such application at line 218 to the discussion.

2. As the fitting of the GMM is a stochastic process, how stable is the choice of one- and two-dimensional models based on the AIC? Also, the number of evaluated topologies seems rather small, especially for the two-dimensional models (from 2x2 to 2x4). Why not go higher?

Because each mixture component would have a full covariance matrix, each mixture would have over 800 associated parameters. We felt that evaluating larger topologies would lead to severe overfitting due to the presence of far more parameters than data points.

3. How do you choose the covariance matrix of each mixture? Are these fully independent from each other, or are some of the coefficients shared?

The covariance matrices are fully independent of each other and are updated independently in each step of our EM algorithm. The covariance matrix would be updated to optimize the log-likelihood of the data.

4. Why do you focus on 95% of most abundant taxa over all three datasets instead per dataset?

This is a conservative evaluation of performance, as it does not take advantage of low-abundance dataset-specific taxa that would allow for distinction between regions in a dataset without computational sophistication. We wanted to be confident that our model was capturing global spatial dynamics in its choice of model dimension instead of just modelling low-abundant taxa.

5. How were the reads preprocessed? At which taxonomic level did you analyze them? If I am not mistaken this is the OTU level, but results for Fig. 6 are displayed on the phylum level. Did you find similar patterns at the OTU level?

Reads were preprocessed in Sheth et. al, 2019. The data provided was only the relative abundances of the taxa. Reads were analyzed at the genus level. In Figure 6, we aggregate the abundances of various genera into a single abundance of their respective phylum (the gray bars). We also show the aggregated abundances of specific families within the Firmicutes phyla because there is some knowledge in science literature of their expected behavior along pH gradients.

6. Are the reads available on a public repository?

The reads from Sheth et. al are available at this github repository: <http://github.com/ravisheth/mapseq>

7. 1241: What is the Inverse-Wishart distribution?

The Inverse-Wishart distribution is a conjugate prior of the multivariate normal distribution when sampling covariance matrices. It is a standard prior used in contemporary Bayesian inference. We now refer to the relevant textbook in the manuscript text (reference 15).

8. Did you use other software packages to implement the method? If so, please reference them.

Qiime2 was used in creating phylogenetic trees (reference 13) and PhILR was used for creating a basis from this tree (reference 12). All other software packages used were standard open-source Python libraries and are incorporated in the environment.yml file of our github repo.

9. Some figures have small captions, which are difficult to read in its current form. See for example the labels in Figures 4 and 5.

We have updated the caption sizes for various figures to avoid this issue.

10. A suggestion (especially as you have quite some space left): as the Materials & Methods section comes in the end, the paper might be more accessible if you repeat some of the

essential elements from it in the Results section. Some things (e.g. the simulation study) only became understandable after reading the paper as a whole. Another thing I would add is that each mixture represents a composition, from which you use the average to inspect changes in composition between different (estimated) locations in the GI.

On lines 97-101, we have added more clarifying language to explain the models being tested. We have also further clarified how we are analyzing the mixtures as you suggested on line 132.

11. The link to the repository with code (and data?) does not seem to work.

This has been fixed - it is now public. We have also added a data availability section pointing to locations of data and the algorithms we developed to analyze them (line 367).

There is a typographical error and a few inconsistencies:

- l22: seem → seems
- Names of datasets are sometimes written with and sometimes without capital letter. Please be consistent.
- Please introduce the meaning of all symbols, such as Q, Q0, ρ , and z_b and upon first mentioning, such as Δ and Δ^* .

The above errors have been corrected.

October 6, 2021

Dr. Itsik Pe'er
500 w 120th st
500 w 120th st
mailcode 0401
mailcode 0401
New York, NY 10027

Re: mSystems00817-21R1 (Directional Gaussian Mixture Models of the gut microbiome elucidate microbial spatial structure)

Dear Dr. Itsik Pe'er:

Your manuscript has been accepted, and I am forwarding it to the ASM Journals Department for publication. For your reference, ASM Journals' address is given below. Before it can be scheduled for publication, your manuscript will be checked by the mSystems senior production editor, Ellie Ghatineh, to make sure that all elements meet the technical requirements for publication. She will contact you if anything needs to be revised before copyediting and production can begin. Otherwise, you will be notified when your proofs are ready to be viewed.

As an open-access publication, mSystems receives no financial support from paid subscriptions and depends on authors' prompt payment of publication fees as soon as their articles are accepted. =

Publication Fees:

We recognize that the video files can become quite large, and so to avoid quality loss ASM suggests sending the video file via <https://www.wetransfer.com/>. When you have a final version of the video and the still ready to share, please send it to Ellie Ghatineh at eghatineh@asmusa.org.

Sincerely,

Vanni Bucci
Editor, mSystems

Journals Department
Phone: 1-202-942-9338